# Selectivity of Transport Processes in Ion-Exchange Membranes: Relationship with the Structure and Methods for Its Improvement

**DOI:** 10.3390/ijms21155517

**Published:** 2020-08-01

**Authors:** Irina Stenina, Daniel Golubenko, Victor Nikonenko, Andrey Yaroslavtsev

**Affiliations:** 1Kurnakov Institute of General and Inorganic Chemistry of the RAS, 119991 Moscow, Russia; stenina@igic.ras.ru (I.S.); xpman2009@yandex.ru (D.G.); 2Membrane Institute, Kuban State University, 350040 Krasnodar, Russia; v_nikonenko@mail.ru

**Keywords:** ion-exchange membranes, functional polymers, selectivity, ionic conductivity, hybrid membranes, grafted membranes

## Abstract

Nowadays, ion-exchange membranes have numerous applications in water desalination, electrolysis, chemistry, food, health, energy, environment and other fields. All of these applications require high selectivity of ion transfer, i.e., high membrane permselectivity. The transport properties of ion-exchange membranes are determined by their structure, composition and preparation method. For various applications, the selectivity of transfer processes can be characterized by different parameters, for example, by the transport number of counterions (permselectivity in electrodialysis) or by the ratio of ionic conductivity to the permeability of some gases (crossover in fuel cells). However, in most cases there is a correlation: the higher the flux density of the target component through the membrane, the lower the selectivity of the process. This correlation has two aspects: first, it follows from the membrane material properties, often expressed as the trade-off between membrane permeability and permselectivity; and, second, it is due to the concentration polarization phenomenon, which increases with an increase in the applied driving force. In this review, both aspects are considered. Recent research and progress in the membrane selectivity improvement, mainly including a number of approaches as crosslinking, nanoparticle doping, surface modification, and the use of special synthetic methods (e.g., synthesis of grafted membranes or membranes with a fairly rigid three-dimensional matrix) are summarized. These approaches are promising for the ion-exchange membranes synthesis for electrodialysis, alternative energy, and the valuable component extraction from natural or waste-water. Perspectives on future development in this research field are also discussed.

## 1. Introduction

Ion exchange membranes (IEMs) are actively used in modern technologies, including water purification, concentration, electrochemical synthesis, and sensors [1,2,3,4,5]. Their use in fuel cells [6,7,8,9], in energy storage and conversion systems, e.g., metal-ion batteries [10,11,12], reverse electrodialysis [13,14,15], and redox batteries [16,17,18,19] has received the most attention in recent years. There are a number of other technologies in which ion-exchange membranes can also be used. For example, they can successfully compete with porous membranes in the processes of alkenes/alkanes separation [20,21,22,23,24]. Undoubtedly, ionic conductivity plays a dominant role in these applications. Even the above-mentioned alkene transfer proceeds via the a complex with silver ions or protons [23,25,26,27].

At the same time, the ion separation by membranes never occurs with 100% selectivity and is usually accompanied by an undesired transfer of molecules or opposite charged ions. Despite the fact that these processes proceed to a much less extent, they usually determine the decrease in the efficiency of electrodialysis, fuel cells and other devices based on ion-exchange membranes [16,28,29,30]. That is why their understanding is so important for numerous membrane technologies.

The selectivity of transport processes in ion-exchange membranes is determined by the structure of their pores and channels [31,32,33,34]. As this structure is formed as a result of the self-organization, it is difficult to control but it is the key aspect which allows one to regulate the ion-exchange membrane selectivity. In addition, there are major differences between the morphology of homogeneous and heterogeneous membranes, which lead to a lower selectivity of the latter [33,35,36]. Nevertheless, lower cost of heterogeneous membranes determines their widespread use in electrodialysis plants and in a number of other devices [37,38].

Recently, increasing attention has been paid to the separation and processing of complex mixtures, including industrial and domestic wastes, sea water [39,40,41,42,43,44,45]. There is every reason to believe that their processing, along with recycling, would determine a major part of production of a number of metals and their compounds [39,46,47,48,49,50]. For the development of such technologies the selectivity between ions with different valences of the same sign (mono- and multivalent) is no less important [51,52,53,54]. The processes of their separation have recently been extensively reviewed [55,56,57,58]. However, this topic cannot be ignored, and in order to avoid overlapping, in this review, we will consider it briefly, focusing on newer studies in this area.

This review is devoted to the consideration of the selectivity of transport processes in ion-exchange membranes. The membrane structure-selectivity relationship and various methods for selectivity management are discussed. The main attention is paid to the selectivity of transport processes of ions with different charges and ions with neutral molecules.

## 2. The Ion Exchange Membrane Structure and Ion Transfer

Description of the ion-exchange membrane structure should be started from one of the benchmark materials—perfluorosulfonic acid membranes of the Nafion type. Nafion is a random copolymer of tetrafluoroethylene and sulfonyl fluoride vinyl ether [59,60]. The first of them forms a hydrophobic perfluorinated backbone, and the second forms a side chain with -SO_3_H (−SO_3_M) terminal group. The latter are hydrophilic and, as result of self-organization, form clusters, which absorb water from the environment. As a result, there is an extended network of pores (4–5 nm in size in a swollen state) and channels filled with water in the membrane. Dissociation of functional groups leads to the formation of an aqueous solution containing dissociated cations [61,62,63,64]. On the contrary, fixed -SO_3_^−^ ions are localized on the pore walls, giving them a negative charge. Due to the electrostatic interaction, most of the cations are localized in a thin Debye layer near the pore walls, and the so-called electrically neutral solution is located in the pore centers (Figure 1) [32,65]. It is generally accepted that the composition of this electrically neutral solution is close to that of the solution contacting with the membrane. When membranes are used in electrodialysis or reverse electrodialysis, the concentration of contacting solution can be rather high [16,31,66]. Moreover, in the central part of the pore, there are a lot of cations and anions which can contribute significantly to the conductivity. On the other hand, it is this pore central part that determines the significant contribution of anionic conductivity, which reduces the selectivity of transport processes. On the contrary, the high field strength in the Debye layer leads to the anion displacement from it, and this layer performs predominantly selective cationic transport [65,67].

When used in fuel cells, the membranes contact with water vapor only. In this case, the electrically neutral solution in their pore centers should be almost pure water with a low cation concentration. This water should freeze near 0 °C. However, due to the small size and high curvature radius, its freezing temperature decreases significantly. Freezing of this water was found in grafted membranes with a very high water uptake and a large pore size [68]. Moreover, its enthalpy value shows that not all water freezes at 0 °C, and about 20 remaining molecules per functional group gradually freeze during further cooling [69]. The water uptake of most ion-exchange membranes is less, therefore, phase transformations are usually not observed near 0 °C.

The water freezing in perfluorinated Nafion membranes, the water uptake of which is usually much less, has been studied in detail. The authors of [60,70] constructed a phase diagram for these membranes at subzero temperatures, from which one can conclude that water freezing near 0 °C is possible for the water uptake of about 20–30 molecules per -SO_3_H group. Mendil-Jakani et al. [71] estimated this water uptake as 50 molecules per functional group. Water in the Nafion membrane pores usually freezes only at subzero temperatures. Most researchers divided this water into bound water, as these water molecules do not freeze due to electrostatic interactions, and free water that have been observed to be freezable [60,72,73,74,75]. However, as the temperature decreases, their ratio changes significantly due to the freezing of water occupying the outer solvating shells of H_3_O^+^ ions [70,72,73]. With a water uptake of four or less per functional group, water freezing is not observed at all [60]. This corresponds to the formation of a stable H_9_O_4_^+^ ion. From the plot of total water vs. free water, Kusoglu et al. [60] estimated amount of non-freezable water as 4.75 molecules per proton of -SO_3_H group. This amount depends substantially on the nature of counterions in the membrane. Some studies revealed that deep cooling of membranes with low water uptake results in glass-transition and desorption of freezing water [71,76].

The higher the water uptake of membranes, the larger the size of pores and channels, which limit ion mobility and conductivity [77,78,79,80] (Figure 2a). These channels are similar to the bottlenecks of transport channels in solid electrolyte. Therefore, the conductivity of ion-exchange membranes increases with increasing water uptake. At the same time, the water uptake increases primarily due to an increase in the volume of the electrically neutral solution in the pore centers (Figure 2b). It is where coions, non-polar and low-polar molecules, e.g., molecules of gases that feed fuel cell, are localized. Since the ion flux is determined by the product of the charge carrier concentration and their diffusion coefficient, the rate of non-selective transfer also increases rapidly with increasing water uptake. Thus, the membrane conductivity improvement is most often accompanied by increase in undesirable transfer of coions and nonpolar molecules that decreases the membrane selectivity [81,82,83,84]. This relationship will be addressed in detail below. This correlation takes place in the case of not only ion-exchange membranes, but also other types of membranes. As indicated by Park et al. [83], all synthetic membranes exhibit a trade-off between permeability/conductivity (the rate of transfer of molecules and ions through a membrane material) and permselectivity (the ability to separate the target component from the feed solution): the higher membrane permeability, the lower permselectivity, and vice versa. For example, the well-known Robeson diagram illustrates this correlation for gas separation membranes [85,86,87,88,89,90].

The reasons for this relationship between water uptake, conductivity, and selectivity of ion-exchange membranes can be seen in Figure 2. With increasing membrane water uptake, the size of pores and channels also increases (Figure 2a). In its turn, an increase in the size of the channels results in an increase in the ion mobility and the ionic conductivity of membranes. On the other hand, there is a significant increase in the coion concentration, which is determined by the fraction of the electroneutral solution in the membrane (Figure 2b). This is the main reason for the increase in the flow of coions and a decrease in the selectivity of ion-exchange membranes with high water uptake. These regularities are typical for both cation-exchange and anion-exchange membranes [91,92,93,94].

It is worth mentioning that the membrane pretreatment at elevated temperatures, pressures as well as the change of the solvent nature (for example, to water-alcohol) also leads to some changes in pore volume, conductivity, and selectivity. These effects were discussed in recent reviews [60,95].

Ion transfer in ion-exchange membranes occurs in solution localized inside the pores, and its mechanism is similar to the mechanism of ion transfer in solutions. For most cations, it includes hopping between different positions, accompanied by the destruction and the reformation of hydration shells. For membranes with low water uptake, some positions in their environment can be occupied by fixed ions (-SO_3_^−^ groups) [96]. For membranes in the hydrogen form, the transfer mechanism changes somewhat and includes a sequence of proton-containing group rotations and proton hops along hydrogen bonds. Moreover, at a high water uptake, cooperative effects play a significant role and proton transfer by the Grotthuss mechanism dominates [97].

The discussed above model for perfluorinated sulfonic acid membrane morphology, is often used for heterogeneous membranes. However, they are usually obtained by hot rolling or pressing of an ion-exchange resin and a plastic binder. Due to this, large macropores remain between their granules [32,36,98,99,100]. Such pores appear between the ion exchanger granules and the plasticizer during the membrane formation as the package defects. As a result, the pore size distribution in heterogeneous membranes is bimodal. The main part of the pores in size (several nm) and structure is close to those for homogeneous membranes, while the other part of the pores is larger (about 1 μm) [35,101]. The transfer through these macropores cannot be selective. Therefore, the heterogeneous membrane selectivity is usually lower than that of homogeneous ones [36,102,103]. Reinforcing mesh used to increase the mechanical strength of heterogeneous membranes can also contribute to a decrease in selectivity [104].

## 3. Pseudo-Homogeneous and Grafted Membranes

The obvious desire is to create relatively cheap membranes, for example, similar in composition to heterogeneous ones, but with high selectivity like in homogeneous membranes, primarily perfluorinated sulfonic acid membranes. To do this, one should get rid of the macropores formed during the heterogeneous membrane preparation. The most evident approach seems to prepare homogeneous membranes based on sulfonated polystyrene (PS), which is the basic ion-exchange material for the preparation of most of heterogeneous membranes. However, it is impossible to obtain films with good mechanical properties from it. That is why a binder, most often polyethylene, is used to form heterogeneous ion-exchange membranes [32,105,106].

From this point of view, it seems logical to use alternative methods for the preparation of IEMs of a similar composition by chemical synthesis. In particular, such a ′′defect-free′′ combination of functionalized PS and hydrophobic polymer can be achieved in the block copolymer synthesis, e.g., IEMs based on triblock styrene–ethylene/butylene–styrene copolymers [107,108]. The well-known Neosepta^®^ membranes are prepared by polymerization of styrene in a paste with polyvinyl chloride particles [109,110].

Another promising approach is the grafted copolymer synthesis based on the radical polymerization of styrene directly inside the hydrophobic film [111,112,113,114]. For example, irradiation of polyethylene with *γ*-rays results in its radical formation. Usually, the lifetime of the radicals is very short and due to their high reactivity, they annihilate very quickly, but in the polymer film they can exist for a long time, at least for several months. The reason for this is the low mobility of polymer chains, which prevents annihilation. This allows one to separate the stages of irradiation and polymerization. By varying the radiation dose and the time of styrene polymerization, it is possible to obtain a material based on the irradiated polyethylene films with different polystyrene content [115,116,117]. In this case, polystyrene is formed as a separate phase in a dense polyethylene film, as if expanding the polyethylene chains without pore formation. Thus, through subsequent sulfonation, ion-exchange membranes with a wide range of ion-exchange capacities and fairly uniform distribution of -SO_3_^−^ groups over the membrane thickness were prepared [115,118,119]. However, membranes with a high degree of grafting and sulfonation swell too much in water, losing their mechanical strength and selectivity. To prevent this, crosslinking with divinylbenzene can be used. Such polymers have a heterogeneous structure, but due to the fact that the heterogeneity size is less than the visible light wavelength, IEMs based on them are transparent. Therefore, they are usually called pseudo-homogeneous. A similar approach was used for a number of polymers [120,121,122,123,124] and allowed authors to obtain materials with desired properties for a number of electromembrane processes [112,125,126].

Radical creation by softer activation, e.g., using ultraviolet radiation is more attractive. However, its energy is not enough to activate polyethylene. For this reason, a polymer containing tertiary carbon atoms, e.g., polymethylpentene, is needed. Its activation allows for the polystyrene grafting with subsequent sulfonation and crosslinking. As a result, a number of cation-exchange membranes (CEMs) with a wide range of ion exchange capacities were prepared [84] as well as anion-exchange membranes (AEMs), which surpass the best commercial membranes in their properties [127].

Styrene polymerization inside stretched films of polyethylene or other polymers can be considered as an alternative approach. Mechanical deformation resulted in formation of multiple small pores inside them, which are then filled with polystyrene [119,128,129,130]. This approach does not prevent the preservation of sufficiently large pores inside the polymer, but in some cases, it is possible to obtain membranes with high selectivity, which can be judged, for example, by their high efficiency in fuel cells.

Depending on the application, membrane materials with high ionic conductivity or selectivity are necessary, but in any case, both of these parameters are important. In particular, for electrodialysis, the ratio of conductivity to cation transfer numbers is most important. The relationship between them can be found from a consideration of the transport of counterions and coions through membranes on the basis of the irreversible thermodynamics [131]. The authors concluded that cation transfer numbers should increase with ionic conductivity increasing. However, due to the reasons discussed in the previous section, their ratio usually turns out to be the reverse.

To compare membrane conductivity and selectivity, a two-dimensional diagram can be used (Figure 3). Membranes with the best transport properties are near the so-called ′′upper bound′′. Homogeneous perfluorosulfonic acid membranes have the best properties among the commercial membranes. They are followed by pseudo-homogeneous and heterogeneous ones. At the same time, the best of the samples of grafted membranes based on polymethylpentene and sulfonated polystyrene match the best perfluorosulfonic acid membranes.

It should be noted that many commercially available ion-exchange membranes used in electromembrane processes, for example, Neosepta^®^, Fumatech^®^, FujiFilm^®^ membranes and others have the pseudo-homogeneous morphology. When designing pseudo-homogeneous IEMs, it is possible to adjust their properties by changing the nature and proportion of the hydrophobic polymer matrix, the degree of functionalization, the nature of the functional group, using reinforcement or crosslinking.

Let us consider how the above factors affect the water uptake of ion-exchange membranes and, hence, their selectivity. Typically, an increase in the proportion of the hydrophobic polymer matrix and its rigidness leads to a decrease in water uptake [84,109,115,133,134,135,136]. For example, Gupta et al. showed that an increase in the grafting degree of sulfonated polystyrene for the radiation grafted FEP-g-polystyrenesulfonic acid membranes from 6 to 40% leads to an increase in water uptake from 2–3 to 63 water molecules per sulfonic-acid group [137]. The authors of [138] demonstrated that an increase in the crystallinity of the cation exchange membranes from 8 to 25% leads to a decrease in their water uptake from 32 to 19 water molecules. It should be noted that the CMX Neosepta^®^ membrane, which made of high rigid polyvinyl chloride, is one of the most mechanically tough and selective commercial IEMs. This is due to the fact that the limitation of the swelling of the hydrophilic ionic domains is accompanied by an increase in their selectivity. Another important parameter is the amount of functional groups [82,136,139,140]. For example, an increase in the sulfonation degree of polystyrene in the cation-exchange membranes based on block copolymers from 12 to 49% leads to an increase in water uptake from 3 to 165 water molecules per sulfonic-acid group.

For anion-exchange membranes, the nature of the functional group represented by various quaternary amines is especially relevant [82,141]. Therefore, Cho et al. [82] showed that AEM based on substituted imidazoles have higher selectivity than those with the trimethylamine functional groups, which was associated with their lower hydration degree. A similar correlation is well known for cation-exchange membranes: during electrodialysis of NaCl solution, Nafion-type membranes with a carboxyl functional group [142,143] are more selective in comparison with more hydrophilic sulfonic acid-based membranes. Another example of ionogenic group nature on swelling properties is CEM with sulfonylimide ionogenic groups showing a high swelling and ionic conductivity in water/organic amide mixed systems [144] compared with sulfonic acid-based membranes.

Ion exchange membranes are often reinforced to improve their selectivity, mechanical properties, or to prevent significant dimensional changes during hydration or drying [145,146,147,148,149]. Polymeric meshes, nonwoven materials, and organic and inorganic fibers are generally used for membrane reinforcement. The reinforcing material often forms directly during the synthesis. Thus, if the membranes are prepared via the polymerization between hot squeezing rollers or plates, the reinforcing mesh controls the thickness of the final film and does not allow the reaction mixture to leak out under pressure [150,151]. In some cases, the monomer polymerization takes place directly in the reinforcing material matrix [119,128,152]. By limiting the swelling of the conductive ionic phase, suitable membrane selectivity can be maintained. However, it is worth noting that high membrane selectivity can only be achieved if reliable adhesion is ensured between the reinforcing and ionic materials and there are no macropores between them. Otherwise, reinforcement will lead to macropore formation and a decrease in selectivity [104].

## 4. Cross-Linking of Polymer Membranes

It can be assumed that there is a certain optimal size of pores and channels connecting them to ensure a good combination of ionic conductivity and selectivity of ion-exchange membranes. On the one hand, to provide high ionic conductivity of the membranes, large channels and pores are required, and hence a high water uptake. However, if we refer to the example with grafted membranes, the fact that membranes with intermediate water uptake have optimal transport properties can be seen [84]. At low water uptake, the whole pore/pore volume is covered by a double electric layer and the coion concentration in them is negligible. Therefore, the selectivity should be very high. But narrow channels and weak pore connection determine too low conductivity of such materials. Even at a high water uptake of the ion-exchange membranes equilibrated with water, the counterion concentration in the inter-pore solution is quite high (about 2–3 mol/L). Therefore, as a first approximation, the Debye layer thickness is assumed to be constant. With increasing water uptake, the pore size increases mainly due to an increase in the volume of the electrically neutral solution. Thus, with water uptake increasing, the counterion transport numbers, corresponding to the membrane selectivity, should decrease.

Another factor determining the conductivity is the charge carrier concentration [97], which is determined by the counterion concentration. Since in most applications, membranes with functional groups based on strong acids and bases are used, it can be assumed that in the hydrated state the dissociation degree of functional groups is close to one and the carrier concentration is determined by the ion-exchange capacity. However, an increase in the concentration of functional groups in the hydrophobic polymer matrix unambiguously leads to an increase in water uptake and, hence, to an increase in the carrier mobility. Since water in the membrane matrix acts as a plasticizer, the elasticity of the matrix, which determines its swelling degree, increases with water uptake increasing. Moreover, a lot of ion-exchange materials with a high ion-exchange capacity lose their mechanical properties or even become water-soluble. This limits their practical application.

In fact, this means that strategies aiming at increasing the membrane conductivity by increasing its ion-exchange capacity or water uptake should definitely decrease its selectivity and the mechanical properties [107,153,154,155]. One of the solutions is the cross-linking of polymer chains. This limits the membrane swelling degree and, hence, leads to an improvement in their selectivity and mechanical properties [156,157,158,159].

In a number of synthetic approaches, cross-linked agents, e.g., divinylbenzene (DVB), can be added to the ionomer during polymerization. Using such methods, highly selective membranes are often obtained. For example, the Neosepta^®^ CMX membrane based on a sulfonated styrene-divinylbenzene copolymer (s-PS-DVB) has apparent cation transport number of 99% (0.5 M/0.1 M NaCl), low permeability for sodium chloride and contains 7–9 water molecules per functional group in the Na^+^-form [160]. Kang et al. [128] showed that an increase in the cross-linking degree for grafted membranes based on sulfonated polystyrene can halve the membrane hydration degree as well as reduce its methanol permeability by 1.8 times. Generally, ion-exchange membrane manufacturing methods, including solvent casting stage, do not imply the possibility of medium or highly cross-linked materials preparation. Cross-linking method requires additional processing steps or the introduction of special groups (reactive sites) [3,161,162,163]. Such approaches are rather relevant for membranes based on polyetheretherketones, polysulfones or block copolymers based on polystyrene and polyolefins. It is worth noting that this can significantly increase the membrane cost.

Most often, divinylbenzene is used as cross-linker in ion-exchange membranes based on styrene copolymers. As shown in [132,164,165], the use of more flexible cross-linkers, for example, bis (vinylphenyl) ethane (Figure 4), allows manufacturing ion-exchange membranes with a better selectivity/conductivity ratio. In particular, at close transfer numbers, the ionic conductivity of membranes crosslinked with bis (vinylphenyl) ethane is four times higher than that of membranes cross-linked by divinylbenzene [132]. This effect is explained by the close reactivity of bis (vinylphenyl) ethane and styrene, which provides more uniform cross-linking and more optimal membrane structure.

At this point, it is also worth considering IEMs based on strongly cross-linked matrices obtained by polymerization of aqueous or amide solutions of ionogenic monomers, such as functionalized acrylamides or styrenes [151,166,167,168]. Well known Fujifilm ion-exchange membranes also belong to this class of materials [104,169]. Due to the high cross-linker content, reaching 60%, a three-dimensional network is formed during the polymerization, which does not swell like ordinary membranes. Therefore, the maximum water uptake of such membranes is limited by the initial solvent volume, which determines the free volume (Figure 5). However, the hydration degree of such membranes is usually more than 18–20 water molecules per one functional group, due to the limited solubility of the monomers and cross-linkers in the reaction mixture. Of course, this to some extent limits the selectivity of such materials. Note that this approach leads to a paradoxical for other types of ion-exchange membranes effect—an increase in the functional group concentration in these membranes leads to a decrease in water uptake per functional group and an increase in selectivity [168,170,171].

## 5. Hybrid Membranes

Another approach to increase the selectivity and decrease the water uptake of ion-exchange membranes is to replace part of the water in the pores by the inorganic particle’s incorporation. Since the membrane pore size is usually small and does not exceed 5 nm [62,172], the size of the introduced particles should also be several nanometers. Doping with inorganic nanoparticles has been used to improve the transport properties of membrane materials since the late 1980s [173,174,175,176,177]. Since organic and inorganic components are simultaneously present in such systems, such membranes are called hybrid membranes. Research in this area was almost simultaneously developed for gas separation membranes, which are often called the Mixed Matrix Membranes [178,179].

In the case of ion-exchange membranes, this approach is more often used to increase their ionic conductivity. For this purpose, nanoparticles of hydrophilic materials, such as silica [180,181,182,183,184,185,186,187], or other oxides [188,189,190,191,192,193,194,195,196,197,198,199] are used. This often leads to an increase in the membrane water uptake, and therefore it is often believed to be the reason for the conductivity increase. However, in a number of hybrid membranes, the conductivity increases despite the lower water uptake. To explain this effect, a model of limited elasticity of the membrane pore walls was proposed [200]. Upon doping, inorganic nanoparticles should displace a part of water molecules from the pore. Since the counterion concentration remains almost the same, this leads to an increase in the osmotic pressure and a pore expansion (Figure 6a,b), accompanied by an expansion of the channels connecting them and, hence, a conductivity increase. However, with increasing nanoparticle size, the elastic forces of the pore walls increase according to Hooke′s law and the osmotic pressure becomes insufficient for their further expansion. Therefore, the membrane water uptake decreases, and new “bottlenecks” that limit conductivity appear in the pores (Figure 6c). When the additive content is above two vol %, the conductivity of the hybrid membranes decreases [180,201,202,203]. This model is confirmed by comparing the data on the ionic conductivity of hybrid membranes with the diffusion coefficients of proton-containing groups determined by NMR with a pulsed magnetic field gradient [201,204]. The authors of [205] developed this model and described with its help the effect of a double electric layer formed around nanoparticles embedded in mesopores and macropores on the electrical conductivity, diffusion permeability of IEMs, and ion transport numbers in them.

Increased conductivity is not the only advantage of hybrid membranes. Some authors also note an improvement in their mechanical properties [177,207,208]. Although it is not always achieved, the mechanical properties of hybrid membranes are usually sufficient for their practical use [16,209].

A noteworthy advantage of hybrid membranes is often their reduced permeability of methanol and other fuel gases [202,210,211,212]. This determines, in particular, the widespread use of hybrid membranes in direct methanol fuel cells (DMFCs) [16,213,214]. A comparison of proton conductivity and methanol permeability for a number of hybrid membranes is given in Table 1.

It is worthwhile to note that gas permeability can be considered as a measure of selectivity of membranes used in fuel cells and the transfer mechanism of non-polar gas molecules or alcohols containing voluminous non-polar fragments is much similar to that of coions. They are also displaced by polar water molecules and cations from the double electric layer and are localized mainly in the electroneutral solution.

The increase in selectivity of hybrid membranes can be explained by the model of limited elasticity of their pore walls (Figure 6). When additive particles are introduced, they displace a part of the electroneutral solution localized in the pore center, thereby decreasing the solubility of gases or alcohols as well as crossover in fuel cells. However, considering the conductivity and selectivity of hybrid membranes, it is necessary to take into account the nature of additive particles or their surface. It has already been noted above that the particle surface should, whenever possible, be hydrophilic for conductivity enhancement. The acid properties of its surface are no less important.

If the additive surface contains strong acid groups, as result of their dissociation, an additional number of counterions (protons, charge carriers) are formed. In this case, the nanoparticle surface acquires a charge of the same sign as the pore walls of cation-exchange membranes. These filler particles should repel from them due to the electrostatic effects, be located in the pore center, and displace only the electrically neutral solution. Moreover, their surface charged negatively will create an additional double electric layer, contributing to the displacement of coions and non-polar molecules as a result of competition with polar water molecules, which interact much more strongly with the charged pore walls. Moreover, such nanoparticles should not decrease the counterion transport along the pore walls, on the contrary, they will contribute to its increase due to an increase in the charge carrier concentration [67,205]. Heteropoly acids, their acid salts with alkali metal cations, or silica particles with heteropoly acids were shown to improve conductivity in such way [218,219,220,221,222,223,224,225]. On the other hand, the surface of such particles is hydrophilic, which contributes to an increase in water uptake. When the composite membranes Nafion/SiO_2_/PWA and Nafion/SiO_2_ were employed as an electrolyte in H_2_/O_2_ PEMFC, the higher current densities (540 and 320 mA/cm^2^ at 0.4 V, respectively) were obtained than that of the Nafion-115 membrane (95 mA/cm^2^), under the operating condition of 110 °C and at the humidified temperature of 100 °C [219].

The opposite situation can be realized if the membrane is doped by nanoparticles with the basic surface [226,227]. In this case, “salt bridges” are formed between the functional groups of the membrane and the basic surface groups, e.g., of the -SO_3_--HNR_3_ type. Their formation decreases the ion-exchange capacity of the membrane and constricts the pore walls, decreasing pore size and water uptake. Despite the fact that the additive surface in this case is also hydrophilic, actually, its basic functional groups interact with the acid groups of the pore walls, reducing their solvation and turns out to be an additional factor that reduces the water uptake and leads to an increase in selectivity of these membranes [228]. In fact, in this case, the role of doping turns out to be similar to that of cross-linking—the membrane conductivity decreases while selectivity increases.

The influence of the dopant surface acidity on the properties of cation-exchange membranes based on polymethylpentene with grafted sulfonated polystyrene has been studied in [229]. ZrO_2_, TiO_2_, and SiO_2_ with increased acidic properties were used as dopants. Their pK_a_ values vary somewhat depending on the preparation method and are 11–12, 8–9, and 5–6, respectively [230]. If the first of them mainly exhibits basic properties, those of the last one are acidic. The ion-exchange capacity and conductivity of hybrid membranes naturally increase in this series of dopants (Table 2). If in the case of zirconia and titania the ion exchange capacity decreases due to the formation of salt bridges (hydrogen ionic bonds of the type -M--O_3_S-), silica does not change it. Based on Poisson-Boltzmann equations, the fraction of negatively charged groups on the surface of a silica nanoparticle at pH = 6 estimated to be in the range of 5–10% [231]. This result is in good agreement with the electrokinetic experimental data reported by Sonnefeld et al. [232]. Dissociation of OH-groups of silica showing weak acid properties is suppressed in the presence of a strong sulfonic acid (sulfonated polystyrene). At the same time, the ionic conductivity of this membrane significantly increases as result of increased pore size and water uptake.

At the same time, the selectivity of ion-exchange membranes significantly decreases in the ZrO_2_-TiO_2_-SiO_2_ series. When zirconia is used as a dopant, the apparent cation transport numbers increase by 7% compared with the initial membrane, for silica it decreases by the same 7%. After the ZrO_2_ introduction, the proton conductivity sharply decreases due to the formation of salt bridges. At the same time, after membrane treatment with 1 M alkali solution, its conductivity increases and reaches almost the same values as for the initial membrane. This change is due to the destruction of salt bridges in the alkaline medium. It should be noted that this effect is reversible—during cyclic treatment with acid and alkali solutions, the conductivity almost reversibly returns to the values of the hydrogen and sodium forms of hybrid membranes, respectively [229].

An increase in the selectivity of transport processes was also noted upon membrane doping with such a weak base as polyaniline [233,234,235,236]. Recently, much attention has been paid to carbon nanomaterials, primarily carbon nanotubes (CNT). They have gained attention due to unique structure and properties such as extraordinary mechanical properties, high surface area, electronic conductivity and chemical stability. Carbon nanotubes are mostly used in fuel cells as a catalytic support for the oxygen reduction reaction [237]. However, carbon nanomaterials have also been used as fillers for fuel cell membranes, e.g., the CNTs introduction can improve their mechanical properties [237,238]. Some authors also reported a decrease in the membrane permeability to methanol as well as its crossover in direct methanol fuel cells with CNTs introduction [237]. The use of carbon nanotubes with a sulfonated surface for this purpose is especially effective, since they can increase the proton conductivity of membranes due to additional charge carrier introduction [239,240,241,242]. Moreover, the negatively charged surface of sulfonated CNTs creates an additional double electric layer around them and increases the membrane selectivity. This leads to a decrease in the methanol permeability of such hybrid membranes and an increase in the fuel cell power [239,243]. For example, a decrease in the methanol permeability of the obtained hybrid membrane by more than 3 times was shown [244]. At the same time, the authors of [245] noted that at very high sulfonation levels a morphological transition caused a decrease in transport properties. There is information about increase in the conductivity of sulfonated poly (etheretherketone) membranes due to their doping with CNTs coated with silica [246] as well as CNTs with imidazole groups on their surface [247,248].

Other carbon materials as dopants for proton-conducting membranes are less studied. However, in recent years, composite materials with graphene have received much more attention [249]. Among the advantages of composite Nafion/graphene membranes, increased proton conductivity is noted [250]. At the same time, the authors of [251] reported a decrease in conductivity and methanol permeability of such membranes. The proton conductivity of sulfonated graphene-Nafion composite membranes at low humidity (20–25% RH) was found to be five times higher than that of a pristine Nafion membrane, while a peak power density of fuel cells based on them is 1.5 times higher [252,253].

## 6. Membranes with Modified Surface

The surface of ion-exchange membranes largely determines their transport properties. Therefore, its modification has long been considered as one of the ways to improve the properties of ion-exchange membranes [32,254]. Various approaches were used for this. The membrane surface profiling is one of the simplest ways. This approach is more effective for modification of heterogeneous membranes [255,256,257,258,259,260]. Most of their surface (75–85% [98,261]) is covered with a plasticizer (polyethylene) film, which is formed during pressing. Only 15–25% of the surface is occupied by ion-exchange resin particles which protrude over the polyethylene film and provide ion transport (Figure 7).

Profiling results in an increase in the active surface area both due to the increase the surface area itself and by reducing the polyethylene coating area. The presence of a profile on the membrane surface significantly reduces both the diffusion path length and the effective thickness of the diffusion layer due to the optimization of hydrodynamic conditions [257,258,259,262,263,264]. The appearance of the tangential component of the electric force acting on the space charge of the solution at the surface of the protrusions results in an increase in electroconvection. During electrodialysis of dilute solutions, this effect leads to an increase in the mass transfer rate through profiled membranes up to 8 times [265].

In a number of studies, plasma treatment was used to modify the surface layer of membranes. In some cases, this leads to an increase in pore size in the surface layer and to a decrease in selectivity [266]. Choi et al., on the contrary, reported that plasma treatment reduces the methanol crossover in Nafion membranes with a simultaneous decrease in proton conductivity as a consequence of the removal of a part of the functional groups [267]. Lee et al. explained a decrease in the methanol crossover of the membranes modified by ion implantation by both the removal of sulfonic acid groups and their substitution by COOH, OO, CO-groups [268]. The coating of a thin fluorocarbon layer on the surface of perforated membranes allowed reducing the wettability of their surface and the methanol permeability by two orders of magnitude [269]. Plasma carbon deposition on the membrane surface led to similar results [270]. It is also reported that, using plasma treatment, aluminosilicate nanotubes can be introduced into the surface layers of membranes [271].

As mentioned above, the properties of heterogeneous membranes are largely determined by the imperfection of their surface, which is usually predominantly coated with a plasticizer, e.g., polyethylene. From this point of view, coating it with a thin layer of polyelectrolyte with better conductivity seems reasonable [272,273]. Perhaps, homogeneous perfluorosulfonic acids of the Nafion^®^ type are one of the optimal materials for coating heterogeneous cation-exchange membranes. Such a coating does not increase the cost of heterogeneous membranes too much, but gets their properties similar to those of homogeneous ones, increasing their conductivity and improving the performance of electromembrane processes at super-limiting currents [274,275]. A perfluorosulfonated polymer film on the surface of a heterogeneous membrane can significantly reduce or even eliminate deposition of sparingly soluble salts on the membrane surface during electrodialysis [276,277]. Of considerable interest is the chemical modification of the AEM surface with a bifunctional polymer solution allowing transforming the functional tertiary and secondary amino groups into the quaternary ones [36,278]. Such a modification can significantly reduce the intensity of undesirable generation of H^+^ and OH^−^ ions near the AEM surface [36] due to a decrease in the concentration of secondary and tertiary amino groups in the membrane surface layer, which exhibit high catalytic activity in this process [279]. An additional improvement in the properties of such membranes can be achieved by modifying the surface coating with inorganic oxide nanoparticles [280]. It can also be noted that membranes with a deposited surface layer can regulate the transfer rates of ions with different charge sizes [56], since multicharged ions predominantly adsorbed in the membrane pores limit the transfer of monovalent ones [281,282].

Bipolar membranes containing two layers (cation-and anion-exchange) with functional groups of the opposite charge are one of the important types of ion-exchange membranes. Due to their high selectivity, they have been successfully used in a number of applications. Very interesting is the recently discovered possibility to convert sunlight into ionic electricity with bipolar membranes [283] and to apply this energy for CO_2_ reduction [284] and for water electrolysis [285,286,287,288,289,290,291,292]. The use of bipolar membranes is also advantageous in reverse electrodialysis [293] and in storage of electrical energy using flow batteries [294]. There is another effective way to improve the selectivity of membranes by coating layers with fixed groups of opposite charge. For example, the surface of cation exchange membranes can be modified with polyethyleneimine [295,296,297,298,299] or polyaniline [234,300,301,302,303]. It should be noted that such membranes also differ in the asymmetry of ion transport and, in particular, in diffusion permeability [304]. This effect is determined by the ion concentration gradient, which complicates their transfer in one of the directions [305]. This phenomenon can be used to improve the selectivity of electrodialysis [306].

Surface modification by deposition of layers containing fixed ions of opposite charge is very often used to increase the selectivity of membranes to mono- and multicharged ions [300,302,306,307,308,309,310]. For this purpose, surface properties such as hydrophilicity, porosity, nature or concentration of functional groups can be changed [311,312,313,314]. However, the layer-by-layer (LBL) modification, described in detail in recent reviews [55,56], increasing the selectivity of transport of ions with different charge sizes, has been considered the most promising. In recent years, this approach has been widely used to create membranes for selective separation of a number of singly and doubly charged ions [309,315,316,317,318,319,320,321,322,323,324]. Their use permits to achieve outstanding separation ratios and developing highly efficient electrodialysis. For example, the modification of Nafion membranes made it possible to achieve a separation coefficient of K^+^ and Mg^2+^ ions in electrodialysis plants exceeding 1000 [320]. However, in this case, the membrane resistance increases by a factor of 20, and the limiting current drops by more than four times. Coating of the layers on the surface of AEM by electric-pulse deposition method increased the separation efficiency of Cl⁻/SO_4_^2^^−^ from 8.93 to 94.4%, while the permselectivity increased from 0.81 to 47. However, a simultaneous increase in membrane resistance should be noted [325]. Nafion^®^ membranes coated with polystyrene sulfate and polyaniline showed a very high selectivity for the electrodialysis separation of Li^+^/Co^2+^ and K^+^/La^3+^ pairs; the separation coefficients exceeded 5000 with a decrease in the limiting current by several hundred times [326]. This approach can also be used to increase the selectivity of separation of ions of the same charge with the use of complexation. Thus, Kazemabad et al. succeeded in increasing the Li^+^/K^+^ selectivity of membranes using polyanionite modified with crown ethers with a multilayer composition [327]. However, due to the availability of recent detailed reviews and limited space in this publication, we will consider only some theoretical aspects of this phenomenon. Experiments and mathematical modelling show that the main role in increasing selectivity is played by the first surface bilayer [316,317,319]. However, the deposition of additional bilayers continues to increase the specific selectivity up to 10 bilayers [319].

## 7. Compromise between the Specific Permselectivity and Permeability. Impact of Concentration Polarization

For the membranes selective for monovalent ions, as well as for other types of membranes [80,81,82,328], there is a compromise between selectivity and the flux of the target component. As Table 3 shows, the higher the density of the ion flux through the membranes, the less specific permselectivity. In particular, the greater the driving force (voltage/current in the case of IEMs or pressure in the case of nanofiltration (NF) membranes), the less specific permselectivity in most cases. In a number of works [52,281,319,320,329,330,331,332,333], the IEM selective permeability significantly decreases with current density increasing and practically disappears when the limiting current density is reached. One of the reasons for the decrease in selective permeability is that the control of mass transfer passes from the membrane to the diffusion layer when approaching the limiting state [281,329,330]. Another reason is water splitting (generation of H^+^ and OH⁻ ions). To increase specific permselectivity, the formation of bi-layer structure on the membrane surface is quite effective [319]. However, the intensity of water splitting increases with an increase of the number of bi-layers because of increasing the number of bipolar junctions [319,320]. This reduces the transfer of the target ion. In addition, the change in pH in the diffusion layer caused by water splitting can affect the properties of the modifying layer [52,320]. Thus, an increase in its swelling and a decrease in charge density can significantly reduce the effect of monovalent permselectivity.

Quite interesting is the ′′odd−even′′ effect of the LBL modified membranes. Abdu et al. [319] describe the case where a CMX cation-exchange membrane was coated by several bilayers of a hyperbranched poly(ethyleneimine) (PEI) bearing positive fixed charges, then by a poly(4-styrenesulfonate) (PSS) layer bearing negative fixed charges (of the same sign as those of the CMX). The specific permselectivity and water splitting were higher in the cases of PEI-terminated bilayers as compared to their values for the PSS-terminated bilayers. Thus, changing the nature of the terminating layer allows switching on and turning off water splitting at the surface of IEMs [7].

To understand the effect of the value of driving force on the membrane specific permselectivity, it is necessary to take into account the current-induced concentration polarization of IEMs in mixed solutions. Mathematical modeling of ion transfer through IEMs in a mixture of several electrolytes was considered in a number of studies [330,332,338,339,340]. For the first time, the dependence of the effective transport numbers of competing ions in a membrane system on the current density was described by Oren and Litan [338]. The authors used the Nernst-Planck equations and the electroneutrality condition written for three types of ions in the membrane and in the diffusion layers adjacent to it. Both diffusion and migration contributions to the ion transport were taken into account in the diffusion layers, and only ion migration in the membrane. A similar model was developed in [330], where both migration and diffusion transport in the membrane were accounted for. It was shown [316,340] that the loss of specific selectivity with increasing current is due to the transition of mass transfer control from the membrane at low currents to the depleted diffusion layer at the currents close to the limiting one. At low currents, the external concentration polarization (deviations of the concentrations from equilibrium in the diffusion layers) is insignificant, and the effective transport numbers of ions are determined by their migration transport numbers in the membrane. With increasing current, the ion concentration in the diffusion layers changes in such a way that the concentration of the selectively transferred counterion (marked as ion 1 in Figure 8) in the depleted diffusion layer decreases much more than the concentration of the competing counterion (ion 2 in Figure 8). Symmetrically, in the enriched diffusion layer, the concentration of the selectively transferred counterion 1 increases more strongly than that of its competitor. As a result, the concentration profiles in the membrane are arranged in such a way that the diffusion flux of the selectively transported counterion is directed opposite to the migration flux, and the diffusion flux of its competitor, on the contrary, is aligned with the migration flux (Figure 8). As a result, concentration polarization, developing with increasing current, restricts the resulting flux of the preferentially transported counterion, and enhances the flux of the competing counterion.

At the limiting current, the mass transfer control passes to the depleted diffusion layer, the concentrations of all ions become zero at the membrane surface facing this diffusion layer. In this approximation, the partial current density of counterion *k*, *i_k_*_lim_, at *i = i*_lim_, can be calculated as follows [330]:(1)$$i_{k\lim}=\frac{F}{\delta }D_{k}z_{k}c_{k}^{o}\left(1+\frac{z_{k}}{z_{3}}\right)$$
where *D_k_*, *z_k_* and *c_k_*^0^ are the diffusion coefficient, charge number and bulk concentration of ion *k* (*k =* 1, 2), respectively; *δ* is the diffusion layer thickness, *F*, the Faraday constant; subscript ′′3′′ refers to the coion, which is common to the considered ternary electrolyte.

There are some papers developing the described models. Fila and Bouzek [332,339] consider also an IEM and two adjacent diffusion layers, but they use the extended Nernst-Planck equation, which contains the convective term. Taking into account the electroosmotic convective transport is important when studying the ion transfer in multi-ionic concentrated solutions, in particular, the application of IEMs in chlor-alkali electrolysis. Geraldes and Afonso [339] applied a linearized form of the Nernst–Planck equations for the same three-layer system with a multicomponent solution. The authors obtained an expression for the limiting current density involving a mass-transfer coefficient and an effective diffusion coefficient of the multi-ionic electrolyte.

The fact that the specific permselectivity depends on the current density makes obvious the problem of optimization of the conditions for the separation of different kinds of ions with the same sign of charge. In the case of a mono-layer membrane, the situation is relatively simple: there are two films, a membrane and a depleted diffusion layer, which can control the mass transfer kinetics. With increasing current, the control passes from the membrane to the diffusion layer. However, in the case of multi-layer membranes, different membrane layers can be limiting; the dependence of the permselectivity on the current density and other experimental conditions may be complicated. These systems seem quite promising but remain poorly studied.

## 8. Conclusions

The above material indicates that the transport properties of ion-exchange membranes are determined primarily by their structure, which in its turn depends on the membrane preparation method. As shown in a number of publications, there is a trade-off between the membrane permeability/conductivity and the selectivity of the transfer rate of the target component. In this regard, the quality of membrane material can be characterized by means of the relation between the permeability and permselectivity, for example, by the relation between the ionic conductivity and counterion transport number (in electrodialysis) or ionic conductivity and permeability of fuel molecules (crossover in fuel cells). To enhance this relation, a wide range of approaches is used, such as crosslinking, preparation of hybrid membranes with embedded nanoparticles, and surface modification. Their use can significantly increase the selectivity of transport processes. For example, cross-linking can significantly reduce coion transport, doping with nanoparticles can increase ionic conductivity or, conversely, selectivity of transport processes. A very interesting approach is the surface modification, which allows one to achieve the separation of mono- and multivalent ions. A group of synthetic methods that make it possible to eliminate the formation of large membrane pores during the synthesis (obtaining grafted membranes or membranes with a fairly rigid three-dimensional matrix) should be separately noted.

The use of these approaches offers opportunity for expanding the prospects of ion-exchange membrane use in such a traditional application as electrodialysis, or in rapidly developing hydrogen energy. On the other hand, surface modification can significantly increase the efficiency of extracting valuable components from natural or waste-water.

## Figures and Tables

**Figure 1 ijms-21-05517-f001:**
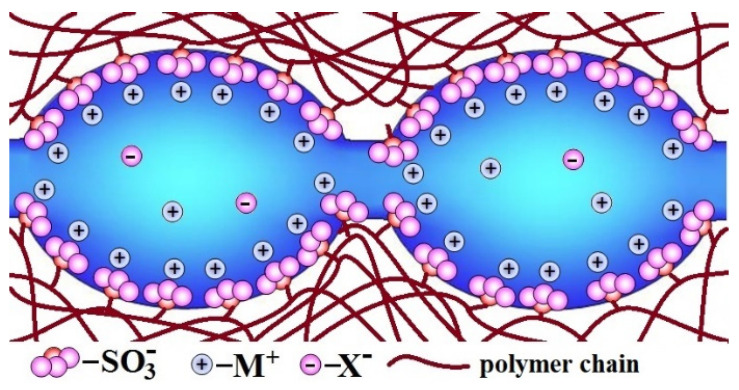
The scheme of pores and channels structure in the ion-exchange membranes (redrawn from [62]).

**Figure 2 ijms-21-05517-f002:**
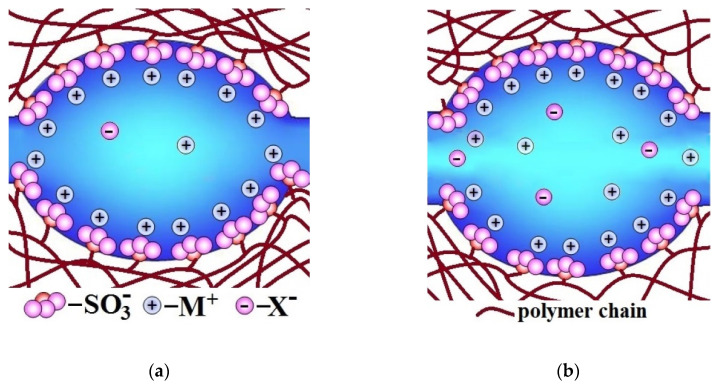
Illustration of the change in pore-channel system of membranes with increasing water uptake (from (**a**) to (**b**)).

**Figure 3 ijms-21-05517-f003:**
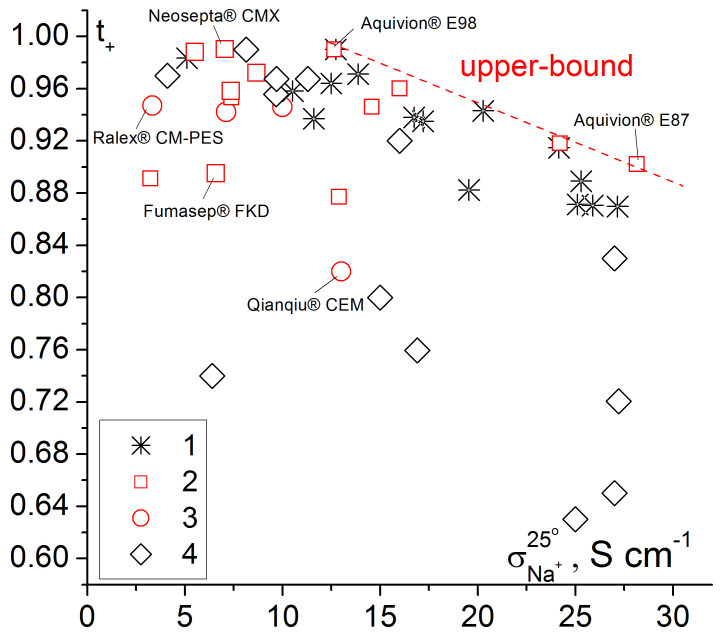
Dependence of apparent transport numbers (0.5/0.1 M NaCl) on the Na^+^ conductivity of various cation exchange membranes. 1—grafted membranes based on UV-oxidized polymethylpentene [84], 2—homogeneous and pseudo-homogeneous materials, 3—heterogeneous membranes, 4—grafted membranes based on polyethylene [115,132]. Some trade names of commercially available membranes are shown in the Figure.

**Figure 4 ijms-21-05517-f004:**
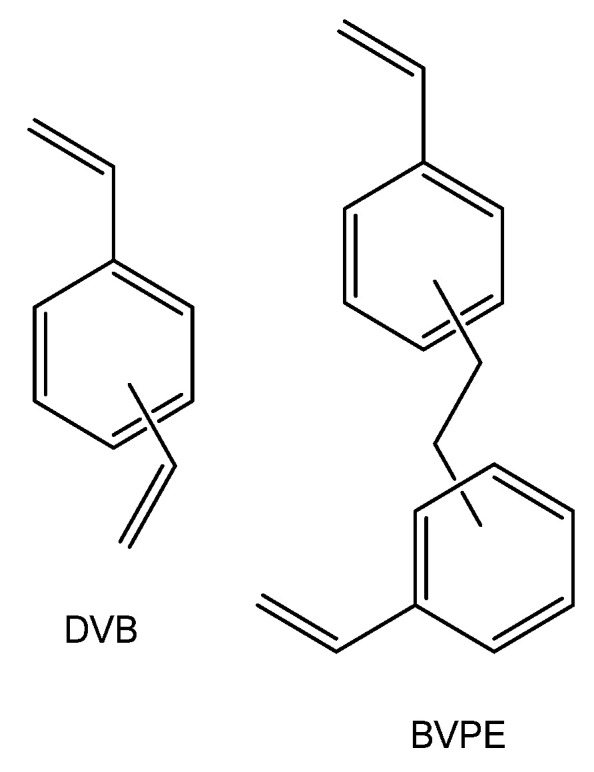
Chemical structures of crosslinking agents: divinylbenzene and bis (vinylphenyl) ethane.

**Figure 5 ijms-21-05517-f005:**
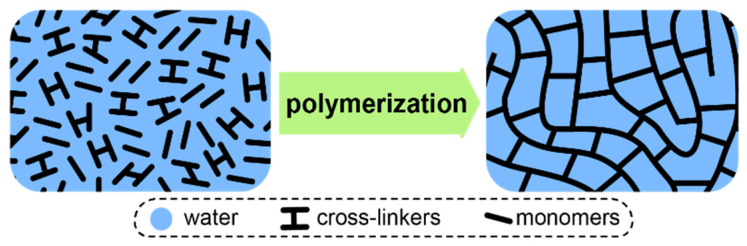
Formation of a typical highly cross-linked membrane via polymerization of aqueous or amide solutions of ionogenic monomers.

**Figure 6 ijms-21-05517-f006:**
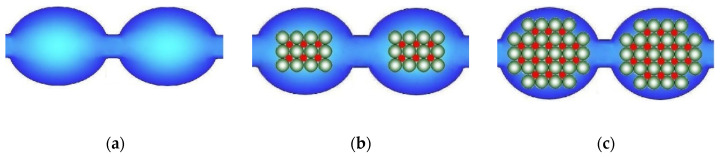
Scheme of the pore structure change: the initial membrane (**a**) and membranes modified with nanoparticles of different sizes (**b**,**c**) according to the case of the model of limited elasticity of the pore walls (redrawn from [206]).

**Figure 7 ijms-21-05517-f007:**
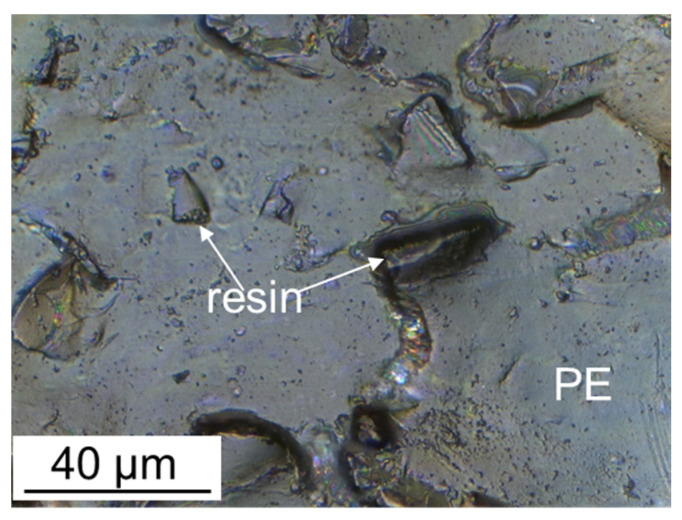
Optical image of the surface of the swollen MA-41 heterogeneous membrane; ′′PE′′ is the polyethylene binder, ′′resin′′ is the ion-exchange resin particle protruding over the polyethylene film (redrawn from [36]).

**Figure 8 ijms-21-05517-f008:**
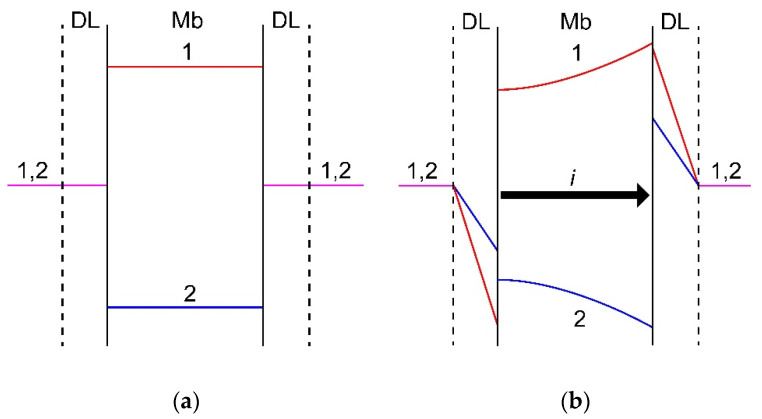
Scheme of concentration profiles in diffusion layers (DL) and a membrane (Mb) established at zero current (**a**) and a current density *i* close to 0.5 *i*_lim_ (**b**). Counterion 1 is preferably sorbed by and transferred through the membrane as compared to counterion 2. The profiles match the results of calculation using model [330]. Explanations are given in the text.

**Table 1 ijms-21-05517-t001:** Comparison of the proton conductivity and methanol permeability of various PEMs for DMFCs application.

Material	Proton Conductivity (S/cm)	Methanol Permeability (10^−7^ cm^2^/s)	Reference
SPAEEKK SPAEEKK/sulfonated silica	0.012 0.043 (20 °C, 100 RH ^1^)	7.83 4.86	[215]
Recast Nafion Recast Nafion/silica	0.043 0.034 (20 °C, 100 RH)	7.53 4.17	[180]
Nafion-117 Sulfonated polyimide Sulfonated polyimide/GO	0.038 0.0021 0.0025 (60 °C, 60 RH)	85.9 20.7 4.31	[216]
Nafion-212 Nafion/SiO_2_/m-BOT (bentonite modified by dodecylamine)	0.0991 0.0667 (35 °C, 60 RH)	1.34 0.25	[184]
Recast Nafion Recast Nafion/aminoacid functionalized SiO_2_ nanofiber	0.075 0.1404 (20 °C, 100 RH)	14 10.2	[187]
SPEEK SPEEK/Aminofunctionalized titania sol	0.0179 0.0624 (20 °C, 100 RH)	6.51 5.82	[217]
Pristine recast Nafion Nafion-silica nanopowder (5 wt%) Nafion-silica MSU-F silica meso-structured cellular foam (0.5 wt%) Nafion-silica MCM-41 (0.25 wt%)	0.051 0.084 0.137 0.100 (30 °C, 100 RH)	6.4 ± 0.1 4.8 ± 0.1 1.4 ± 0.1 2.5 ± 0.1	[183]

^1^ RH—relative humidity.

**Table 2 ijms-21-05517-t002:** Ionic conductivities and apparent transport numbers of some grafted hybrid CEM in Na^+^ form with the dopant content of 8–10 wt% [229].

Property	Initial Membrane	Dopant		
		ZrO_2_	TiO_2_	SiO_2_
IEC mg eq/g	2.1	1.1	1.9	2.1
*ω* (H_2_O), wt%	50	29	45	55
σ_Na+_, mS cm^−1^ (in 0.5 M NaCl)	25.3	10.1	22.3	31.6
t_+app_ ^1^	88	95	88	81

^1^ apparent transport number measured between 0.5 M and 0.1 M NaCl solutions.

**Table 3 ijms-21-05517-t003:** The estimations of the specific permselectivity and ionic flux densities for various membranes.

Membrane	Flux Density, j, mol·h^−1^·m^−2^	Selectivity	Driving Force	Reference
Biological membrane	j(K^+^)—6	K^+^/Na^+^ > 1000	Diffusion	[334]
Membrane 1 (NF-270)	j(K^+^) = 2.5	K^+^/Mg^2+^ < 2	Pressure 28 bar	[335]
Membrane 2 (LLC)	j(K^+^) = 1.3×10^−4^	K^+^/Mg^2+^—33	Pressure 28 bar	[335]
Track-etched membrane PET Lumirror^®^	j(K^+^) = 13.9 in 1 M NaCl	K^+^/Mg^2+^ = 135	Voltage 10 V	[336]
Zwitterionic Polyelectrolyte (SBQAPPO)	j(Na^+^) = 0.67 in 0.1 M NaCl + 0.1 M MgCl_2_	Na^+^/Mg^2+^ = 7.4, H^+^/Zn^2+^ = 23.5	Current density 14 mA cm^−2^	[337]
Membrane MK-40	j(Na^+^) ≈ 100 in 1 M NaCl	No specific selectivity	Voltage 0.5 V, *i* = *i*_lim_ = 250 mA·cm^−2^	unpublished data

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
