# Peer review of "Selectivity of Transport Processes in Ion-Exchange Membranes: Relationship with the Structure and Methods for Its Improvement"

_ijms, 2020, doi:10.3390/ijms21155517_

Round 1
Reviewer 1 Report
The manuscript was about the impact of ion-exchange membranes structure on their transport properties. The manuscript could be interesting to the readers of the International Journal of Molecular Sciences. However, some revisions are needed to improve the manuscript. And after reviewing the manuscript, comments are shown below:
- The transport properties of ion-exchange membranes are determined primarily by their structure, which in turn depends on the membrane preparation method. More discussion on the impact of membrane preparation processes, including choice of solvent and processing temperatures on their ion selectivity is missing
- Line 132-133, “With increasing membrane water uptake, the size of pores and channels also increases.” More discussion on how to control the size of the ionic channel of Anion-exchange membrane should be provided.
- The transport mechanism of ions should be included in the manuscript.
Author Response
Dear Reviewer,
Thank you very much for carefully reading the manuscript and making useful comments. We highly appreciate your comments. All of them are taken into account. The necessary changes have been made to the manuscript, they are marked with a yellow marker. Please, find below our responses.
The manuscript was about the impact of ion-exchange membranes structure on their transport properties. The manuscript could be interesting to the readers of the International Journal of Molecular Sciences. However, some revisions are needed to improve the manuscript. And after reviewing the manuscript, comments are shown below:
The transport properties of ion-exchange membranes are determined primarily by their structure, which in turn depends on the membrane preparation method. More discussion on the impact of membrane preparation processes, including choice of solvent and processing temperatures on their ion selectivity is missing
Response: The reviewer is absolutely right. The choice of solvent and the treatment temperature of membranes also have a significant effect on their ionic conductivity and selectivity. However, the processes considered in this review are associated with the filling of membrane pores with an aqueous solution. As for the methods and temperature of membrane processing, this was the topic of our recent review [Safronova, E.Y., Stenina, I.A., Yaroslavtsev, A.B./ The possibility of changing the transport properties of ion-exchange membranes by their treatment.// Petroleum Chemistry. 2017. V. 57 (4), p. 299-305]. But the additional discussion about this topic can rather confuse the reader and complicate the perception of the material, especially since these effects are largely determined by the nature of the membrane matrix, the degree of its crosslinking, and exchange capacity. We considered it possible only to mention the influence of treatment.
Line 132-133, “With increasing membrane water uptake, the size of pores and channels also increases.” More discussion on how to control the size of the ionic channel of anion-exchange membrane should be provided.
Response: In accordance with these comments, additional information concerning anion exchange membranes is included in the text of the article.
The transport mechanism of ions should be included in the manuscript.
Response: Additional information on the ion transport mechanism is included in the manuscript.
Authors
Reviewer 2 Report
The review "Selectivity of transport processes in ion-exchange membranes: relationship with the structure and methods for its improvement" by Stenina et al reports work done on improvement of membrane selectivity and membrane transport of ion-exchange membranes.
Authors have referred to up to date references in literature besides references backdated to the last two decades. all sections are well written and void of grammatical errors. Therefore, I belive this review will appeal to readers of IJMS.
Author Response
Dear Reviewer,
Thank you very much for carefully reading the manuscript and its high appreciation.
Best regards,
Authors
Reviewer 3 Report
The manuscript authored by I. Stenina et al. presents an up-to-date review in such a topic of considerable interest of the academy and industry communities as the interplay of the structure of ion exchange membranes and their functional characteristics. The authors itemize their study in eight separate sections, including Introduction and Conclusions. They overview a great deal of modern scholarly works in this field, focusing mainly on the trade-off of the permeability and selectivity in relation to the membrane microstructure. Starting from the very basic information on the ion exchange membrane structure and homogeneous membranes, they describe approaches to improve the ion exchange membrane functionality by grafting, cross-linking, mixing with additives, and surface processing.
One of the sections (Section 7) is devoted to discussing the immanent compromise between permeability and permselectivity through voltage- (current-) induced concentration polarization, which has been a matter of a number of models and simulations reviewed by the authors. Unfortunately, this section is too speculative and contains too little graphic information (sketches/drawings, graphs, equations etc.), which considerably complicates its comprehension by the reader who lacks expertise within this area.
Overall, this work is a useful and solid overview of contemporary studies on ion exchange membranes. However, I have several comments, mostly related to the language deficiencies, that could help improve the paper.
48,167,179,296,481,526 Please correct using the construction "allow + verb"
69 A verb is missing.
98 "in detail" is more appropriate.
Please consider whether a definite or indefinite article is needed in L. 100, 146,408,411-412,423,428-429,519,551,557.
112-113: "the size... that limits ion mobility and conductivity [77–80] (Fig.2a)." - The reason for the limitation is not clear from the figure.
143 "and another part of the pores has larger size (about 1 μm)" - Where do these pores come from?
150 "membranes primarily" - A comma is likely missing.
151 "seems to be to prepare" - Something is wrong with this sentence.
153 "of most of" is likely the correct wording.
164 "-rays" - Which rays?
How does the sentence in 219-220 cohere with that in 220-223?
232 "anion-exchange membranes" - Please provide the abbreviation
262 Passive voice is expected
284 Either a singular "number" or no indefinite article is expected.
285,Table 2 "(0.5 M/0.1 M NaCl)" - Please explain (at one instance) what this notation means.
293,362 Please correct in "worth to note" either to "worthwhile" or to "noting".
345 "its help"?
402-403 Please reword for a proper conjunction use.
421-422,491 Please reword.
424 "returns"
452,456 The sentences seem unfinished.
480 Something is wrong with the meaning of the end of this sentence.
499-500 The sentence seems awkward.
510,528,531,534,537 Please respect the proper sub-/superscripting.
550 What is NF?
561 The first and the last lines in Table 3 are the same.
598 There is no typed delta in the explanation to formula (1).
Author Response
Dear Reviewer,
Thank you very much for carefully reading the manuscript and making useful comments. We highly appreciate your comments. All of them are taken into account. The necessary changes have been made to the manuscript, they are marked with a yellow marker. Please, find below our responses.
One of the sections (Section 7) is devoted to discussing the immanent compromise between permeability and permselectivity through voltage- (current-) induced concentration polarization, which has been a matter of a number of models and simulations reviewed by the authors. Unfortunately, this section is too speculative and contains too little graphic information (sketches/drawings, graphs, equations etc.), which considerably complicates its comprehension by the reader who lacks expertise within this area.
Response: Thank you for this suggestion. We agree with you, an additional figure (Fig. 8, page 16) was added to the manuscript.
Overall, this work is a useful and solid overview of contemporary studies on ion exchange membranes. However, I have several comments, mostly related to the language deficiencies, that could help improve the paper.
48,167,179,296,481,526 Please correct using the construction "allow + verb"
69 A verb is missing.
98 "in detail" is more appropriate.
Please consider whether a definite or indefinite article is needed in L. 100, 146,408,411-412,423,428-429,519,551,557.
112-113: "the size... that limits ion mobility and conductivity [77–80] (Fig.2a)." - The reason for the limitation is not clear from the figure.
143 "and another part of the pores has larger size (about 1 μm)" - Where do these pores come from?
150 "membranes primarily" - A comma is likely missing.
151 "seems to be to prepare" - Something is wrong with this sentence.
153 "of most of" is likely the correct wording.
164 "-rays" - Which rays?
How does the sentence in 219-220 cohere with that in 220-223?
232 "anion-exchange membranes" - Please provide the abbreviation
262 Passive voice is expected
284 Either a singular "number" or no indefinite article is expected.
285,Table 2 "(0.5 M/0.1 M NaCl)" - Please explain (at one instance) what this notation means.
293,362 Please correct in "worth to note" either to "worthwhile" or to "noting".
345 "its help"?
402-403 Please reword for a proper conjunction use.
421-422,491 Please reword.
424 "returns"
452,456 The sentences seem unfinished.
480 Something is wrong with the meaning of the end of this sentence.
499-500 The sentence seems awkward.
510,528,531,534,537 Please respect the proper sub-/superscripting.
550 What is NF?
561 The first and the last lines in Table 3 are the same.
598 There is no typed delta in the explanation to formula (1).
Response: Many thanks for these comments. All of them are taken into account and the necessary changes have been made.